# Sol2Vy: Leveraging Solidity-Trained Models for Vulnerability Detection in Vyper Smart Contracts

## Abstract

Smart contracts have transformed decentralized finance, but flaws in their logic still create major security threats and financial losses. Most existing vulnerability detection techniques focus on well-supported languages like Solidity, while low-resource counterparts such as Vyper remain largely underexplored due to scarce analysis tools and limited labeled datasets. To address this gap, we introduce **Sol2Vy**, a novel framework that enables vulnerability detection in Vyper smart contracts using models trained solely on Solidity. The key enabler is SlithIR, which we leverage as an intermediate representation to achieve effective cross-language knowledge transfer. Our framework follows a principled three-stage design that integrates unsupervised knowledge learning, supervised vulnerability detection model training on Solidity, and cross-language testing on Vyper. This approach eliminates the need for extensive labeled Vyper datasets typically required to build an accurate vulnerability detection model. We implement and evaluate **Sol2Vy** on three critical vulnerability types: reentrancy, weak randomness, and unchecked transfer. Experimental results show that **Sol2Vy** achieves strong detection performance on Vyper contracts despite being trained exclusively on Solidity, significantly outperforming existing tools.

## 1 Introduction

Smart contracts have become the foundation of decentralized applications and finance, yet their vulnerabilities continue to cause severe security and financial incidents. Detecting these vulnerabilities is therefore of paramount importance. Existing approaches fall into two categories. *The first relies on traditional program analysis techniques*, such as Mythril Sharma & Sharma (2022), Slither Feist et al. (2019), and Securify Tsankov et al. (2018). However, static analysis tools depend heavily on manually crafted patterns and often suffer from high false positive rates, especially on newer compiler versions, while fuzzing-based dynamic approaches struggle with path explosion, limiting their practical applicability. *The second category leverages deep learning*, which offers greater scalability and accuracy by automatically learning semantic features from large datasets.

Existing deep learning-based approaches, however, overwhelmingly target Solidity, the dominant Ethereum smart contract language, which benefits from large, labeled datasets with well-curated vulnerability samples to support the training of robust detection models. In contrast, Vyper remains severely underexplored. Vyper is a rising alternative with Python-like syntax and a security-oriented design that emphasizes simplicity, readability, and a reduced attack surface, making it attractive to developers who prioritize correctness and safety. It is well known that training deep learning models requires a sufficient amount of labeled data. However, *due to the scarcity of labeled vulnerability datasets in Vyper, to date, no deep learning-based models exist for Vyper vulnerability detection.*
**Our Idea and Approach.** In this work, we take *the first step toward developing deep learning-based vulnerability detection for Vyper smart contracts*. To overcome the challenge of data scarcity, our idea is to *reuse a model trained on Solidity to detect vulnerabilities in Vyper*. Achieving such cross-language knowledge transfer, however, is non-trivial due to the syntactic and semantic differences between the two languages. Our **key observation** is that SlithIR Feist et al. (2019), a language-agnostic intermediate representation, can serve as a bridge between Solidity and Vyper.

SlithIR captures smart contract semantics in a simplified, transferable form, enabling transferable representation learning across languages.

Building on this insight, we propose a novel three-stage framework, named **Sol2Vy**. Both Solidity and Vyper contracts are lifted to SlithIR using Slither. The *first stage* is *unsupervised transferable knowledge learning*, where we design a multi-modal architecture that captures both sequential and hierarchical representations of SlithIR. We train this architecture by minimizing the Maximum Mean Discrepancy (MMD) loss between the two languages, thereby learning *transferable, language-agnostic knowledge* from unlabeled Solidity and Vyper datasets. The *second stage* is *supervised vulnerability detection in Solidity*. Here, we add a classification network on top of the multi-modal architecture and train it using labeled vulnerable and safe *Solidity* contracts. During this stage, only the classification network is trained, while the parameters of the multi-modal architecture are frozen to preserve the transferable knowledge. The *third stage* is *testing on Vyper*, where the trained multi-modal architecture and classification network are applied directly to Vyper contracts *without any modification. Notably, **Sol2Vy** requires no labeled Vyper vulnerable samples for training, yet it is still capable of effectively detecting vulnerabilities in Vyper contracts.*

**Results.** We implement our approach, **Sol2Vy**, to extend transferable vulnerability detection capabilities from Solidity to Vyper. When training and testing on Solidity, we achieve the AUC scores of 0.90, 0.91 and 0.90 on reentrancy (RE), weak randomness (WR) and unchecked transfer (UT). When we train it on Solidity and test it on Vyper, **Sol2Vy** achieves AUC scores of 0.87, 0.88 and 0.86. The decreased AUC scores are minimal: 0.03, 0.03 and 0.04 for RE, WR and UT, indicating good transferable knowledge learning to support our idea. The results achieved by **Sol2Vy** also outperform baseline methods of Slither Feist et al. (2019) and Mythril, demonstrating effective vulnerability detection capabilities on Vyper. Below we highlight our contributions:

- We propose **Sol2Vy**, a novel framework that extends vulnerability detection capabilities from Solidity to Vyper, enabling the use of abundant labeled datasets in Solidity for effective model training.

- We lift Solidity and Vyper smart contracts to SlithIR, which is a transferable representation for Ethereum smart contract languages. We design a multi-modal encoder architecture to encode both sequential and hierarchical representations of the SlithIR to capture the language-agnostic semantics.

- We train the multi-modal architecture to learn transferable language-agnostic knowledge by minimizing the MMD loss between Solidity and Vyper representations. We reuse the trained model that preserves the learned transferable knowledge for downstream vulnerability detection model training.

- The training of **Sol2Vy** does not require any labeled Vyper smart contracts, yet the model is still capable of detecting vulnerable smart contracts in Vyper effectively.

## 2 RELATED WORK

Blockchain represents a transformative paradigm in distributed computing Perez & Livshits (2021); Sharma et al. (2023); Chen et al. (2020); Albert et al. (2018); Chen et al. (2025a); Wu et al. (2025b); Yi et al. (2022); Jin et al. (2022); Ruggiero et al. (2024). Solidity remains the dominant language for EVM, while Vyper provides a more Pythonic alternative. Despite the maturity of vulnerability detection tools for Solidity, analysis support for low-resource languages like Vyper remains limited.

Vulnerability detection in smart contracts has progressed through multiple approaches. Traditional static approaches include Slither Feist et al. (2019) and Smartcheck Tikhomirov et al. (2018). Dynamic analysis has evolved through tools like Echidna Grieco et al. (2020), Mythril Sharma & Sharma (2022), ILF He et al. (2019), RLF Su et al. (2022), xFuzz Xue et al. (2022), CrossFuzz Yang et al. (2024), sFuzz Nguyen et al. (2020), SMARTIAN Choi et al. (2021), SCFuzzer Wu et al. (2024), FunFuzz Ye et al. (2024) and Harvey Wüstholz & Christakis (2020). However the support of traditional methods on Vyper is quite limited. For example, Slither and Mythril relies on manually crafted patterns or rules to detect vulnerabilities in Vyper, which is of low coverage and Recently, deep learning models have gained much contributions He et al. (2024a; 2025); He (2024); Wei & Vasconcelos (2023); Wang et al. (2024; 2023); Ahmad & Luo (2024); Zeng & Luo (2025); Wu et al. (2025a); Cui et al. (2022a); Wu & Arafin (2025); Zhang et al. (2024); Zeng et al. (2025a); Sun et al. (2025); Jiang et al.; Li et al. (2025a); Liu et al. (2024a); Xu et al. (2025a;b); Hou et al. (2025); Hu

et al. (2025); Ma et al. (2025); Li et al. (2025b); Chen et al. (2025b); Zhou (2025b); Ge et al. (2024); Xie et al. (2024a;b); Zhou (2025a); He et al. (2024b); Zeng et al. (2025b); Lin et al. (2025); Wang et al. (2025); Lan et al. (2025b;a); Boi et al. (2024); Liu et al. (2024b); Sendner et al. (2023); So et al. (2021); Liu et al. (2021); Hu et al. (2023); Chen et al. (2024); Sun et al. (2024); Cui et al. (2022b); Smolka et al. (2023). Deep learning models often require large labeled datasets, which are scarce in low-resource languages like Vyper. This uneven distribution of analysis tools across languages creates a significant security gap. To mitigate this gap, we propose *a novel SlithIR-based framework that learns transferable language-agnostic knowledge across Solidity and Vyper*, enabling the utilization of Solidity's abundant dataset.

## 3 OVERVIEW

### 3.1 THREE STAGES, TWO TYPES OF DATASETS

`Sol2Vy` aims to develop a robust deep learning model that can extend vulnerability detection capabilities from Solidity to Vyper. Our approach includes three stages: (1) Unsupervised transferable knowledge learning on unlabeled Solidity and Vyper contracts, (2) Supervised vulnerability detection training on labeled Solidity contracts and (3) Testing on Vyper labeled contracts.

We define two types of datasets.
**General Dataset.** This dataset is used to learn transferable language-agnostic knowledge shared by Solidity and Vyper in the first stage. It contains *unlabeled* Solidity and Vyper smart contracts.
**Vulnerability Detection Dataset.** This dataset is related to the vulnerability detection task. A vulnerability detection *Solidity* dataset contains a set of safe contracts and contracts labeled with certain vulnerabilities.

Note that by *data scarcity*, we specifically refer to the *vulnerability detection dataset*. For the *general dataset*, we can easily collect numerous smart contracts. For the vulnerability detection dataset, there are abundant samples for *Solidity*, given its predominant status in blockchain. The vulnerability detection datasets for Vyper are scarce due to its relatively recent emergence in the web3 ecosystem and lack of mature security analysis tooling. Hence, *directly training a vulnerability detection model on Vyper is infeasible*. We therefore propose leveraging a model trained on *Solidity* vulnerability detection datasets to test *Vyper* smart contracts for vulnerability detection.

### 3.2 WHY SLITHIR?

SlithIR is a language-agnostic intermediate representation that abstracts away much of the surface-level syntax of Ethereum smart contract languages. Both Solidity and Vyper can be lifted into SlithIR using the Slither framework, which produces a standardized three-address code format that captures core semantics and control flow. This makes SlithIR a natural candidate for cross-language vulnerability detection.

However, directly relying on SlithIR is not sufficient. We observe that semantically equivalent source code in Solidity and Vyper does not always yield perfectly aligned SlithIR after lifting (see Appendix B). The discrepancies arise from, but are not limited to, two factors: (1) *Compilation-specific variations*: `solc` and `vyperlang` generates distinct intermediate representations even for semantically equivalent code (e.g., Solidity uses `MODIFIER_CALL` to handle function modifiers, while Vyper inlines these checks within function bodies) and (2) *Control Flow Representation*: Given some unique high-level mechanism in Solidity (i.e., inheritance, overloading), the two languages generate divergent abstract syntax trees and SlithIR for the same piece of program. As a result, in our preliminary experiments, a model trained solely on SlithIR lifted from Solidity contracts performed substantially worse when applied directly to SlithIR lifted from Vyper contracts (see "Ablation study" in Section 5.4).

To bridge this gap between Solidity and Vyper SlithIR, we design an unsupervised pipeline that explicitly minimizes the distributional distance between their SlithIR corpora. By training a multi-modal encoder with Maximum Mean Discrepancy (MMD) loss on large unlabeled corpora from both languages, we learn transferable, language-agnostic representations that capture transferable semantics. This transferable representation then forms the foundation of our downstream vulnerability detection model, enabling a classifier trained on Solidity to analyze Vyper contracts without requiring labeled Vyper vulnerability data.

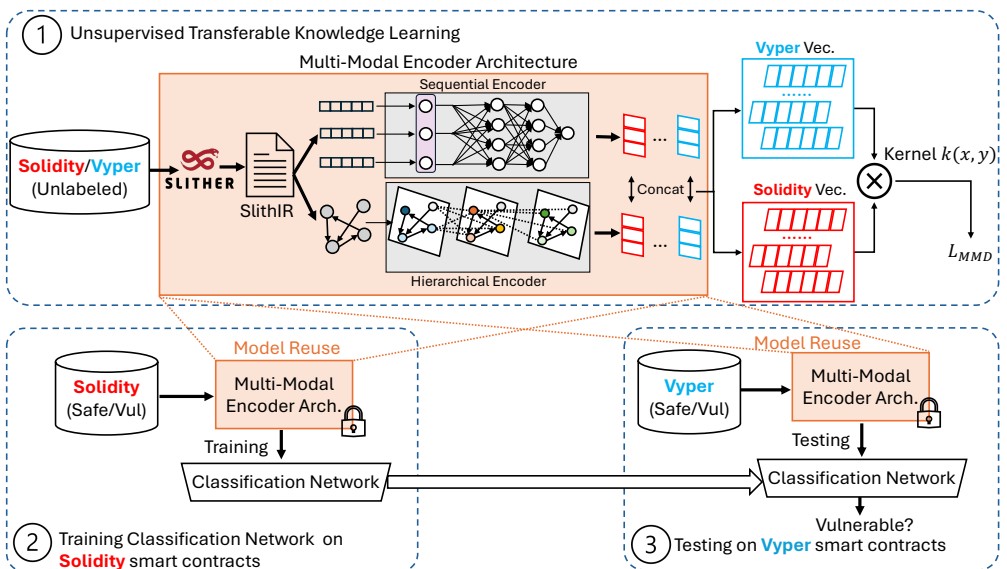

Figure 1: Applying **Sol2Vy** to detect vulnerability in Vyper smart contracts by learning transferable knowledge from Solidity &Vyper corpus and training a classification network on Solidity.

We also consider EVM bytecode; however, as detailed in Appendix A, significant variations in compiled output make it an unreliable solution for our purpose.

## 4 MODEL DESIGN

### 4.1 MODEL ARCHITECTURE

Figure 1 shows an overview of **Sol2Vy**. Step ① is the ***unsupervised transferable knowledge learning stage***. We train the sequential and hierarchical encoders to learn transferable language-invariant knowledge using *unlabeled* Solidity and Vyper datasets. Step ② is the ***supervised vulnerability detection training stage***. In this stage, we include *labeled* safe and vulnerable *Solidity* datasets to train a classification network for vulnerability detection. Step ③ is the ***testing stage***. We use the trained multi-modal encoders and classification network to test on Vyper smart contracts *without any modification*. Note there is *no overlap* between the datasets used for training in *all three stages*.

### 4.2 IR PROCESSING

**IR Extraction.** The first step in our pipeline is to lift both Solidity and Vyper source code into SlithIR. SlithIR abstracts away the syntactic differences of the two languages, providing a standardized, three-address code format that captures the core semantics and control flow. We first extract the Abstract Syntax Trees (ASTs) through the compilers (`solc` for Solidity and `Vyperlang` for Vyper). Then we employ Slither to construct the transferable SlithIR from the ASTs.

**IR Tokenization.** We decompose each SlithIR instruction into constituent components based on operation types, operands, and control flow structures. The tokenization process transforms the three-address code format of SlithIR into discrete tokens while preserving semantic information critical for vulnerability detection: (1) Operation Tokenization: Core operations (e.g., `ASSIGN`, `CONDITION`) are preserved as atomic tokens to maintain instruction semantics. (2) Variable Tokenization: State variables, local variables, and temporary variables (e.g., `TMP_0`, `TMP_1`) are tokenized separately to distinguish their scope and lifetime. (3) Type Information: Type annotations (e.g., `uint256`, `bool`, `address`) are retained as separate tokens to preserve type-safety semantics.

### 4.3 MULTI-MODAL IR FEATURE ENCODING

To fully capture the sequential and structural semantic information inside SlithIR, we design a multi-modal framework to encode *both the sequential and structural* architecture of SlithIR. The multi-

modal encoders consists of two components that operate in parallel on the same SlithIR input: (1) **Sequential Encoder** processes SlithIR instructions as a sequence to capture temporal dependencies and execution flow patterns (2) **Hierarchical Encoder** processes the Abstract Syntax Tree (AST) derived from SlithIR to capture structural relationships and architectural patterns.

**Sequential Encoder.** We employ a Transformer-based architecture for the sequential encoder, leveraging its proven effectiveness in modeling long-range dependencies in sequential data. The tokenized and processed SlithIR instructions are fed to the sequential encoder to generate embeddings that encode both the operation semantics and the associated metadata.

**Hierarchical Encoder.** The hierarchical encoder operates on the Abstract Syntax Tree (AST) representation derived from the SlithIR. We construct a graph representation $G = (V, E)$ where nodes $V$ correspond to syntactic elements such as functions, control flow constructs, variable declarations, and expression components. The edges $E$ in this graph represent structural relationships including parent-child relationships in the AST, data flow dependencies, and control flow connections.

To process this graph structure, we employ Graph Attention Networks (GATs), which provide the capability to learn adaptive attention weights for different types of structural relationships. The GAT layers aggregate information from neighboring nodes through learned attention weights, enabling the propagation of contextual information throughout the hierarchical structure. This aggregation process captures multi-hop relationships in the graph, allowing the model to understand complex structural dependencies that may span multiple levels of the syntax tree.

## 4.4 Unsupervised transferable Knowledge Learning

The first stage focuses exclusively on learning transferable knowledge that is invariant across Solidity and Vyper while preserving the semantic richness necessary for downstream vulnerability detection. During this stage, we train only the sequential and hierarchical encoders, *without involving any classification components or vulnerability levels*.

**Feature Extraction and Combination.** For each smart contract (Solidity/Vyper), we lift it to SlithIR and process it (see Section 4.2). Then we feed the processed SlithIR through our multi-modal architecture (see Section 4.3). The sequential encoder processes the tokenized SlithIR instructions and outputs a feature vector $h_{\text{seq}} \in \mathbb{R}^{d_{\text{seq}}}$. The hierarchical encoder processes the AST-derived graph structure and outputs a feature vector $h_{\text{hie}} \in \mathbb{R}^{d_{\text{hie}}}$. We concatenate these presentations to form the final feature vector: $z = [h_{\text{seq}}, h_{\text{hie}}]$ (see step ① in Figure 1, where $d = d_{\text{seq}} + d_{\text{hie}}$ is the total dimensionality of the combined feature representation.

**Maximum Mean Discrepancy (MMD) Loss.** To achieve transferable knowledge learning, we employ Maximum Mean Discrepancy (MMD) to minimize the distributional distance between Solidity and Vyper feature representations. MMD is a non-parametric statistical method that measures the distance between two probability distributions by comparing their mean embeddings in a Reproducing Kernel Hilbert Space (RKHS). Specifically, given a batch of feature vectors from Soidity $(z_1^S, z_2^S, \ldots, z_{n_S}^S)$ and Vyper $(z_1^V, z_2^V, \ldots, z_{n_V}^V)$. The MMD loss $\mathcal{L}_{\text{MMD}}$ is calculated in Equation (1).

$$\mathcal{L}_{\text{MMD}} = \left\| \frac{1}{n_S} \sum_{i=1}^{n_S} \phi(\mathbf{z}_i^S) - \frac{1}{n_V} \sum_{j=1}^{n_V} \phi(\mathbf{z}_j^V) \right\|_{\mathcal{H}}^2 \tag{1}$$

where $\phi(\cdot) : \mathbb{R}^d \to \mathcal{H}$ is the feature mapping function that maps input features into the RKHS $\mathcal{H}$,

**Kernel Design and Implementation.** The feature mapping function $\phi(\cdot)$ is implicitly defined through the kernel trick using a combination of linear and Gaussian Radial Basis Function (RBF) kernels. We use a composite kernel defined in Equation (2).

$$k(x, y) = \alpha k_{\text{Linear}}(x, y) + (1 - \alpha) k_{\text{RBF}}(x, y) \tag{2}$$

where $k_{\text{Linear}}(x, y) = x^T y$ captures linear relationships between features. And $k_{\text{RBF}}(x, y) = \exp(-\gamma \|(x - y)^2\|)$ captures non-linear similarities. $\alpha$ is the bandwidth hyperparameter that balances the contribution of linear and non-linear components. In practice, we use Equation (3) to calculate $\mathcal{L}_{\text{MMD}}$ to avoid explicit computation of the feature mapping $\phi(\cdot)$ while still measuring the distance between language distributions in RKHS $\mathcal{H}$. This transferable knowledge learning stage operates in an unsupervised manner, *requiring no labeled Vyper data*.

$$\mathcal{L}_{\text{MMD}} = \frac{1}{n_S^2} \sum_{i=1}^{n_S} \sum_{i'=1}^{n_S} k(\mathbf{z}_i^S, \mathbf{z}_{i'}^S) + \frac{1}{n_V^2} \sum_{j=1}^{n_V} \sum_{j'=1}^{n_V} k(\mathbf{z}_j^V, \mathbf{z}_{j'}^V) - \frac{2}{n_S n_V} \sum_{i=1}^{n_S} \sum_{j=1}^{n_V} k(\mathbf{z}_i^S, \mathbf{z}_j^V) \qquad (3)$$

### 4.5 VULNERABILITY DETECTION

Once the multi-modal encoders are trained to produce language-agnostic representations, we proceed to the vulnerability detection task, leveraging the learned transferable knowledge. This process involves the following stages: supervised training on Solidity, and cross-language testing on Vyper. **Training on Solidity.** In this stage, the weights of the sequential and hierarchical encoders from Stage ① are frozen to preserve the learned transferable knowledge. A classification network is added on top of the frozen encoders for vulnerability detection. We train the network using labeled Solidity smart contracts (safe and vulnerable), handling each vulnerability type separately.

The classification network consists of a multi-layer perceptron (MLP). It takes the concatenated feature vector $z = [h_{\text{seq}}, h_{\text{hie}}] \in \mathbb{R}^d$ from the frozen encoders as input, where $d = d_{\text{seq}} + d_{\text{hie}}$ represents the combined dimensionality. We employ two fully connected hidden layers with ReLU activation functions. The final output layer produces vulnerability predictions using a sigmoid function for binary classification (vulnerable or safe).
**Testing on Vyper.** After training, the model can be applied directly to Vyper smart contracts. The multi-modal encoders and classification network process each contract to predict whether it is safe or vulnerable, without any additional fine-tuning or modification.

## 5 EVALUATION

### 5.1 EXPERIMENTAL SETUP & EVALUATION METRIC

We implement `Sol2Vy` using Transformer and Graph Attention Network. We conduct experiments to assess `Sol2Vy`'s vulnerability detection effectiveness. Additionally, we perform a hyperparameter study on the kernel composition and few-shot analysis with labeled Vyper samples. For evaluation metrics, we report both Area Under Curve (AUC) and F-1 score for each vulnerability type.

### 5.2 UNSUPERVISED TRANSFERABLE KNOWLEDGE LEARNING
**General Dataset.** We collect Solidity and Vyper smart contracts from diverse and authoritative sources, including Etherscan Etherscan and GitHub repositories github. We collect 1842 Solidity contracts and 1193 Vyper contracts. The general dataset is used for learning language-invariant transferable knowledge. We set 80% of the dataset for training and 20% for validation. It is important to note that the general dataset used to learn transferable language-agnostic knowledge has *no overlap* with the vulnerability detection dataset used for the vulnerability detection task.
**Learning Transferable Knowledge.** We train the sequential and structural encoder by minimizing the MMD loss between Solidity and Vyper. To ensure stable convergence, we employ a curriculum learning strategy where we gradually increase the complexity of the unlabeled contract dataset throughout training. This approach helps the model learn fundamental transferable knowledge before tackling more challenging distributional differences. The training stops until the validation MMD loss drops below 0.2.

### 5.3 SUPERVISED VULNERABILITY DETECTION
**Vulnerability Detection Training Dataset in *Solidity*.** We build this dataset from multiple popular smart contract repositories like SmartBugs smartbugs (2021), SWC-registry smartbugs (2020), DAppSCAN DAppSCAN (2023), Etherscan Etherscan. Since there are some inconsistent or incorrect labels in existing datasets, we implement a systematic label verification process to ensure data quality. For initial vulnerability labeling, we employ multiple established tools, including Slither Feist et al. (2019), Securify Tsankov et al. (2018), Mythril Sharma & Sharma (2022), Echidna Grieco et al. (2020), which provide automated detection capabilities for known vulnerability patterns. We apply a consensus criterion where vulnerabilities must be identified by at least two independent tools. Then, a manual verification is conducted to guarantee the correctness of the labels. Any contract with a disputed label is excluded from the dataset to maintain label reliability.

Table 1: Vulnerability detection results

| | | RE | | WR | | UT | |
|---|---|---|---|---|---|---|---|
| | | AUC | F-1 | AUC | F-1 | AUC | F-1 |
| Traditional Methods | Slither (Static Method) | 0.54 | 0.62 | 0.57 | 0.58 | 0.59 | 0.60 |
| | Mythril (Dynamic Method) | 0.64 | 0.67 | 0.67 | 0.65 | 0.62 | 0.68 |
| Deep Learning Based Method | Same-language Model (Train & Test on *Vyper*) | 0.78 | 0.80 | 0.73 | 0.77 | 0.77 | 0.79 |
| *Our Method* | **Sol2Vy** (Train on *Solidity* & Test on *Vyper*) | **0.87** | **0.85** | **0.88** | **0.86** | **0.86** | **0.85** |

We collect 1275, 753, 785 and 813 Solidity contracts that are labeled as safe, reentrancy (RE), weak randomness (WR) and unchecked transfer (UT), respectively. We use $80\%$ for training and $20\%$ for validation. Note that the vulnerability detection training dataset *has no overlap* with the unlabeled general dataset used for the transferable knowledge learning stage.

**Training Classification Layer on Solidity.** We freeze the sequential and structural encoders and append a classification layer to the combined output. The classification network is then trained on the Solidity vulnerability detection dataset until the validation loss falls below 0.5.

**Vulnerability Detection Dataset in *Vyper*.** Due to the limited availability of existing labeled Vyper datasets, we construct a vulnerability detection *Vyper* dataset that *contains the same vulnerability types* through systematic collection and expert annotation. To ensure vulnerability coverage, we systematically create vulnerable contract variants by introducing the three target vulnerability patterns into safe contract templates. For labeling, we apply the same rigorous verification protocol used to construct task-specific Solidity dataset. Each Vyper contract undergoes analysis through Slither and Mythril (both support Vyper), followed by a detailed manual code review by smart contract security experts. We collect 434, 236, 226 and 307 Vyper contracts that are labeled as safe, reentrancy (RE), weak randomness (WR) and unchecked transfer (UT), respectively.

**Detection Results on Vyper.** We use the model trained on Solidity to test Vyper smart contract for vulnerability detection. For a fair comparison with the same-language model (see Section 5.4), we use $20\%$ of the Vyper test samples for evaluation in this experiment. Results using the full Vyper test set are provided in Appendix C. The testing results are shown in Table 1. We achieve AUC scores of 0.87, 0.88 and 0.86 for reentrancy (RE), weak randomness (WR) and unchecked transfer (UT). The corresponding F-1 scores are also satisfactory: 0.85, 0.86 and 0.85 for RE, WR and UT.

When few-shot training is applied (i.e., using only a small number of Vyper vulnerable samples for training) the AUC improves substantially reaching at least 0.91 (see the few-shot analysis in Section 5.6). Collecting a limited number of Vyper vulnerable samples, e.g., around 10, is feasible, and we use only these samples for few-shot fine-tuning. This approach avoids the need to collect a large number of Vyper vulnerable samples while still achieving significant performance gains.

## 5.4 MODEL COMPARISON

**Baselines.** Existing approaches for detecting vulnerabilities in Vyper smart contracts are limited to traditional program analysis techniques. Accordingly, we consider two representative baselines: Slither Feist et al. (2019), a static analysis tool that extracts ASTs and detects vulnerabilities through manually crafted pattern matching, and Mythril Sharma & Sharma (2022), a dynamic analysis framework that employs symbolic execution on EVM bytecode lifted from Vyper contracts.

Currently, no deep learning–based models target vulnerability detection in Vyper. To complement our evaluation, we therefore construct a deep learning baseline by training our multi-modal encoder architecture with a classification network directly on SlithIR lifted from Vyper smart contracts, without transferable knowledge learning or parameter freezing. We refer to this configuration as the *same-language model* (train and test on Vyper). We split the Vyper vulnerability detection dataset into $80\%$ for training and $20\%$ for testing. The results of the two traditional baselines and this same-language deep learning model are reported in Table 1.

**Comparison with Slither.** Our approach demonstrates substantial improvements over Slither across all three vulnerability types. Take reentrancy as an example, **Sol2Vy** achieves an AUC of 0.87 compared to Slither's 0.54, representing a $61\%$ improvement. This significant performance gap stems from the *pattern-matching limitation* in Slither's approach when applied to Vyper contracts. Slither relies heavily on manually crafted syntactic patterns, which fail in Vyper due to the language's dif-

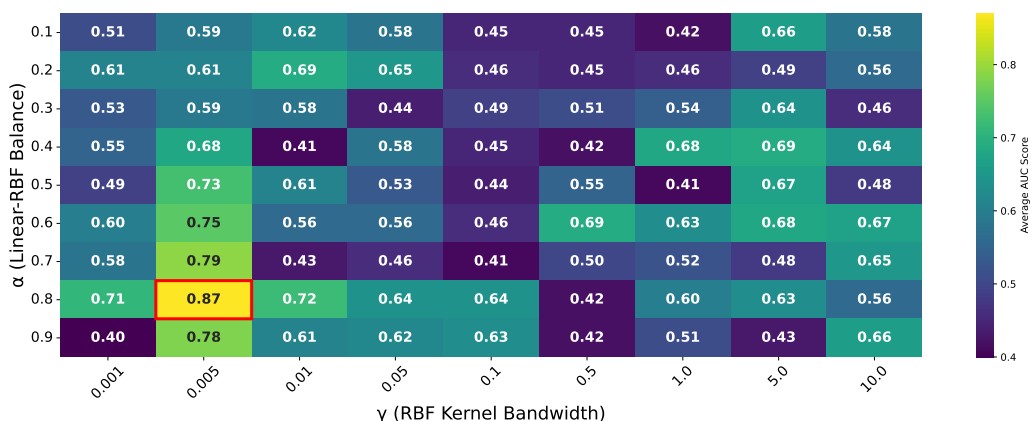

Figure 2: Heatmap showing the average AUC value when varying the hyperparameters $\alpha$ and $\gamma$.

ferent syntactic structures and programming paradigms. Our SlithIR-based approach `Sol2Vy` abstracts away these syntactic differences, enabling pattern recognition at the semantic level.

**Comparison with Mythril.** While Mythril performs better than Slither due to its bytecode-level analysis approach, `Sol2Vy` still outperforms it significantly. For weak randomness, our method achieves 0.88 AUC versus Mythril's 0.67. There are *two critical limitations* for Mythril: (1) Mythril faces severe path explosion problems when analyzing Vyper contracts due to the language's different control flow patterns and the compiler's code generation strategies. Our empirical analysis shows that Mythril times out on 31% of complex Vyper contracts ($\geq 300$ lines). While `Sol2Vy` completes analysis of all test contracts within reasonable time bounds. (2) The compilation to bytecode *loses high-level semantic information that our SlithIR-based approach preserves*. This loss particularly impacts the detection of business logic vulnerabilities (e.g., reentrancy) that depend on high-level logic rather than execution paths.

**Comparison with Same-Language Model.** As we compare the results to the *same-language model* trained and tested *solely* on Vyper, we only use 20% of the vulnerability detection *Vyper* dataset for testing `Sol2Vy`. We also evaluate the performance when testing on *all* the vulnerability detection *Vyper* dataset in Appendix C. When we compare the results of `Sol2Vy` to the same-language model, we observe that our model outperforms the same-language model for all types of vulnerabilities. The same-language model performs worse due to the data scarcity problem: the available Vyper training data is insufficient to train a robust model. However, collecting a large number of Vyper vulnerable samples is challenging, which motivated our approach of leveraging knowledge learned from Solidity and applying the Solidity-trained model to Vyper smart contracts.

**Ablation Study: Without Learning Transferable Knowledge.** To assess the impact of learning transferable knowledge, we perform an ablation study to isolate the effect of this phase. When we train on Solidity and test on Vyper without the transferable knowledge learning stage, the AUC scores drop to 0.71 for RE, 0.70 for WR, and 0.69 for UT. These results underscore the crucial role of transferable knowledge learning in our framework.

## 5.5 HYPERPARAMETER STUDY: KERNEL COMPOSITION

**Experimental Setup.** To understand the impact of kernel composition on transferable knowledge learning effectiveness, we conduct a hyperparameter study focusing on $\alpha$ and $\gamma$ parameters in Equation 2. We systematically vary $\alpha$ and $\gamma$ to evaluate their combined effect on vulnerability detection performance. For each parameter combination, we train `Sol2Vy` using the same three-stage approach and evaluate on all three vulnerability types (RE, WR, UT) using the Vyper test set.

**Result Analysis.** Figure 2 presents the average AUC scores across three vulnerability types for different hyperparameter combinations. We observe that the optimal combination is $\alpha = 0.8$ and $\gamma = 0.005$, achieving an average AUC of 0.87. This configuration can be explained in two aspects: (1) *High linear component emphasis:* The optimal configuration heavily favors the linear kernel component ($\alpha = 0.8$), suggesting that the domain gap between Solidity and Vyper is primarily characterized by linear transformations rather than complex nonlinear mappings. This indicates that while the two languages have syntactic differences, their semantic structures maintain largely linear relationships in the representation space. (2) *Fine-grained RBF Bandwidth:* The very small $\gamma$ value

Table 2: Few-shot analysis results (S = safe Vyper contracts; V = vulnerable Vyper contracts).

| | | 0-shot | 4-shot | | | 8-shot | | | 12-shot | | | 16-shot | | | 20-shot | | |
|---|---|---|---|---|---|---|---|---|---|---|---|---|---|---|---|---|---|
| | | (base) | 4S | 4V | 2S+2V | 8S | 8V | 4S+4V | 12S | 12V | 6S+6V | 16S | 16V | 8S+8V | 20S | 20V | 10S+10V |
| RE | AUC | 0.87 | 0.88 | 0.88 | 0.89 | 0.89 | 0.88 | 0.92 | 0.94 | 0.93 | **0.95** | 0.91 | 0.90 | 0.94 | 0.93 | 0.91 | 0.94 |
| | F-1 | 0.85 | 0.86 | 0.87 | 0.88 | 0.88 | 0.87 | 0.91 | 0.93 | 0.93 | **0.94** | 0.90 | 0.89 | 0.93 | 0.92 | 0.90 | 0.93 |
| WR | AUC | 0.88 | 0.89 | 0.89 | 0.90 | 0.90 | 0.91 | 0.92 | 0.92 | 0.91 | **0.93** | 0.89 | 0.89 | 0.92 | 0.91 | 0.89 | 0.92 |
| | F-1 | 0.86 | 0.87 | 0.87 | 0.88 | 0.87 | 0.88 | 0.89 | 0.90 | 0.91 | **0.92** | 0.88 | 0.87 | 0.91 | 0.90 | 0.88 | 0.92 |
| UT | AUC | 0.86 | 0.87 | 0.87 | 0.88 | 0.88 | 0.89 | 0.90 | 0.92 | 0.91 | **0.94** | 0.87 | 0.89 | 0.92 | 0.89 | 0.91 | 0.92 |
| | F-1 | 0.85 | 0.86 | 0.86 | 0.87 | 0.87 | 0.88 | 0.89 | 0.93 | 0.92 | **0.94** | 0.88 | 0.87 | 0.93 | 0.90 | 0.89 | 0.92 |

(0.005) captures long-range semantic similarities while being robust to local syntactic variations. This bandwidth setting ensures that semantically similar code patterns across languages are mapped to nearby points in feature space. Larger $\gamma$ values (over 0.1) create overly narrow kernels that fragment the feature space and fail to generalize across language boundaries.

**Sensitivity Analysis.** The performance landscape exhibits significant variability, indicating high sensitivity to parameter selection. (1) $\alpha$ *sensitivity:* Performance shows a sharp peak at $\alpha = 0.8$, with substantial degradation at both higher ($\alpha = 0.9$) and lower ($\alpha \leq 0.7$) values. This suggests a critical threshold where linear alignment mechanisms become dominant for effective cross-language transfer. (2) $\gamma$ *sensitivity* The optimal $\gamma = 0.005$ represents a narrow sweet spot. Both smaller ($\gamma = 0.001$) and larger ($\gamma \geq 0.01$) values show decreased performance, indicating that $\gamma$ must be precisely tuned to capture the appropriate scale of semantic similarities.

### 5.6 FEW-SHOT ANALYSIS

**Experimental Setup.** To investigate the potential benefits of incorporating limited Vyper training data, we conduct a few-shot analysis exploring how small amounts of labeled Vyper contracts affect `Sol2Vy`'s performance when added to the Solidity training dataset. This analysis addresses the scenario where practitioners might have access to a small number of labeled Vyper contracts and wish to understand whether including them in training provides improvements.

We systematically evaluate fifteen different few-shot configurations by adding varying numbers and compositions of Vyper contracts to the vulnerability detection Solidity training dataset. The configurations span from 4-shot to 20-shot scenarios with three composition strategies: (1) **Safe-only additions**: Adding only safe Vyper contracts (4S, 8S, 12S, 16S, 20S), (2) **Vulnerable-only additions**: Adding only vulnerable Vyper contracts (4V, 8V, 12V, 16V, 20V), and (3) **Balanced additions**: Adding equal numbers of safe and vulnerable contracts (2S+2V, 4S+4V, 6S+6V, 8S+8V, 10S+10V). Note that the added Vyper smart contracts *have no overlap* with the Vyper contracts for testing. For each configuration, we retrain the classification network while maintaining the same frozen multi-modal encoder architecture from the unsupervised transferable knowledge learning stage.

**Result Analysis.** Table 2 presents the few-shot analysis results across all three vulnerability types. Several key patterns emerge from our findings: (1) **Progressive Performance Gains**: Adding small amounts of Vyper training data consistently improves performance across all vulnerability types. The gains are most pronounced when moving from zero-shot to few-shot scenarios, with diminishing returns as more contracts are added. (2) **Saturation Effects**: Performance improvements exhibit strong saturation effects beyond 16-20 examples, suggesting that `Sol2Vy` can effectively leverage small amounts of target-domain data without requiring extensive labeled datasets. (3) **Composition Sensitivity**: The composition of few-shot examples significantly impacts performance outcomes, with balanced datasets containing both safe and vulnerable contracts consistently outperforming imbalanced datasets. For instance, adding 4S+4V contracts yields superior performance compared to adding 8S or 8V contracts. This pattern holds across all shot levels, demonstrating that diversity in few-shot examples is more valuable than raw quantity.

## 6 CONCLUSION

In this work, we address the critical security gap in smart contract vulnerability detection for low-resource languages Vyper by introducing `Sol2Vy`, a novel transferable knowledge learning framework for vulnerability detection in Vyper. Our approach leverages SlithIR as an intermediate representation to bridge the domain gap between Solidity and Vyper, enabling effective vulnerability detection on Vyper contracts using only labeled Solidity training data.

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

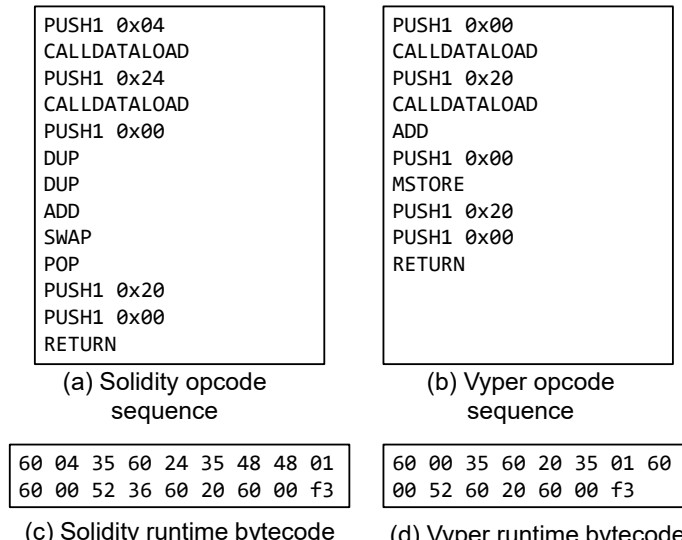

Figure 3: Opcode sequences and runtime bytecode for the `sum()` function in Solidity and Vyper.

## A  BYTECODE DISCUSSION

Intuitively, since both Solidity and Vyper ultimately compile to EVM bytecode for execution, a tool designed to analyze Solidity bytecode should also be applicable to Vyper bytecode. This approach faces two significant challenges: (1) **Bytecode variation:** Solidity and Vyper use separate compilers to generate bytecode with variations. Furthermore, different compiler versions, language-specific abstractions and optimization settings can produce widely varying bytecode for identical source code. This variability complicates attempts to develop universal bytecode-level detection patterns Chen et al. (2020); Albert et al. (2018). (2) **Loss of Semantic Information:** High-level semantic vulnerabilities, including business logic flaws and access control issues, depend on understanding naming conventions, code organization patterns, and implicit developer intentions that compilation obscures Tsankov et al. (2018); Luu et al. (2016). These semantic nuances, which are essential for accurate vulnerability detection, are better preserved in source code.

Furthermore, to demonstrate why a direct bytecode-level approach is problematic, we conducted an experiment comparing the compiled outputs from Solidity and Vyper. We implemented an identical Counter smart contract in both languages, each containing a simple function to sum two input parameters. We use the `Solc 0.8.0` and `Vyperlang 0.3.0` compilers, respectively. Figure 3 presents the compilation outputs, including the opcode sequence and runtime bytecode. When we compare the bytecode and opcode sequences, we observe that they are not strictly identical. This shows that relying on bytecode for comparison is not applicable for Vyper.

## B  DEMONSTRATIONS OF SLITHIR

In Figure 4, we give a demonstration of SlithIR lifted from both Solidity and Vyper using a semantic equivalent `mint()` implementation. The source code for Solidity and Vyper smart contract are shown in Figure 4(a) and Figure 4(b). By comparing the statements in Figure 4(c) and Figure 4(d), we can see the SlithIRs look very different given the semantic equivalent function implementation. So, SlithIR is not a bridge and requires transferable knowledge learning before training a vulnerability detection model.

## C  VULNERABILITY DETECTION USING ALL VYPER SMART CONTRACTS

In Table 1, we give the vulnerability detection results using 20% of task-specific Vyper datasets with comparing with the Same-language Model. In this section, we give the vulnerability detection

```
function mint(address to, uint256 amount)
external payable onlyWhitelisted mintingCooldown
require(amount > 0, "Amount must be greater than 0");
require(msg.value >= mintingFee, "Insufficient fee");
lastMintTime[msg.sender] = block.timestamp;
mintCount[msg.sender]++;
emit TokensMinted(to, amount, mintingFee);
```

(a) **Solidity** source code

```
def mint(_to: address, _amount: uint256):

assert _amount > 0, "Amount must be greater than 0"
assert msg.value >= self.minting_fee, "Insufficient fee"
self.last_mint_time[msg.sender]= block.timestamp
self.mint_count[msg.sender] += 1
log TokensMinted(_to, _amount, self.minting_fee)
```

(b) **Vyper** source code

```
Function Contract.mint(address,uint256)
TMP_0(bool) = amount_1 > 0
TMP_1(None) = SOLIDITY_CALL
require(bool,string)(TMP_0,"Amount must be greater than 0")
TMP_2(bool) = msg.value >= mintingFee_4
TMP_3(None) = SOLIDITY_CALL
require(bool,string)(TMP_2,"Insufficient fee")
REF_0(uint256) -> lastMintTime_0[msg.sender]
REF_0(uint256) := block.timestamp(uint256)
REF_1(uint256) -> mintCount_3[msg.sender]
REF_1 = REF_1 + 1
Emit TokensMinted(to_1,amount_1,mintingFee_4)
MODIFIER_CALL, TokenContractSimple.onlyWhitelisted()()
MODIFIER_CALL, TokenContractSimple.mintingCooldown()()
```

(c) **Solidity** SlithIR statements

```
def mint(address,uint256)
TMP_16(bool) = _amount_1 > 0
TMP_17(None) = INTERNAL_CALL
require(bool,string)(TMP_16,"Amount must be greater than 0")
TMP_21(bool) = msg.value >= minting_fee_2
TMP_22(None) = INTERNAL_CALL
require(bool,string)(TMP_21,"Insufficient fee")
REF_3(bool) -> whitelisted_addresses_1[msg.sender]
TMP_15(None) = INTERNAL_CALL
require(bool,string)(REF_3,"Not whitelisted")
REF_4(uint256) -> last_mint_time_1[msg.sender]
TMP_23(uint256) = REF_4 + 3600
TMP_24(bool) = block.timestamp >= TMP_23
TMP_25(None) = INTERNAL_CALL
require(bool,string)(TMP_24,"Minting cooldown active")
REF_5(uint256) -> last_mint_time_1[msg.sender]
REF_5(uint256) := block.timestamp(uint256)
REF_6(uint256) -> mint_count_2[msg.sender]
REF_6 = REF_6 + 1
Emit TokensMinted(_to_1,_amount_1,minting_fee_3)
```

(d) **Vyper** SlithIR statements

Figure 4: Comparison of SlithIR from Solidity &Vyper of `mint()` function

Table 3: Vulnerability detection results using *all* Vyper contracts

| | | RE | | WR | | UT | |
|---|---|---|---|---|---|---|---|
| | | AUC | F-1 | AUC | F-1 | AUC | F-1 |
| Traditional Methods | Slither (Static Method) | 0.57 | 0.63 | 0.60 | 0.61 | 0.63 | 0.62 |
| | Mythril (Dynamic Method) | 0.66 | 0.70 | 0.71 | 0.67 | 0.66 | 0.70 |
| Deep Learning Based Method | Same-language Model (Train & Test on *Vyper*) | - | - | - | - | - | - |
| *Our Method* | **Sol2Vy** (Train on *Solidity* & Test on *Vyper*) | **0.87** | **0.85** | **0.89** | **0.85** | **0.88** | **0.84** |

results using *all* the task-specific Vyper datasets *without* comparing with the Same-language model. The results are shown in Table 3.