# OpenReview forum: "Sol2Vy : Leveraging Solidity-Trained Models for Vyper Smart Contract Analysis"
_ICLR.cc/2026/Conference — ICLR 2026 Conference Withdrawn Submission_

### Official Review · Reviewer_ngZJ · 2025-10-21

**Soundness:** 2
**Presentation:** 3
**Contribution:** 2
**Rating:** 4
**Confidence:** 4

**Summary:**

This paper proposes Sol2Vy, a framework for cross-language smart contract vulnerability detection. The key idea is to leverage abundant labeled Solidity contracts to detect vulnerabilities in Vyper contracts. This method employs SlithIR, a language-agnostic intermediate representation, and uses an unsupervised MMD-based alignment to learn transferable representations between Solidity and Vyper, followed by a supervised vulnerability detection phase trained only on Solidity data. Experimental results show that Sol2Vy achieves strong performance on three vulnerability types (reentrancy, unchecked transfer, and weak randomness), outperforming static tools and same-language baselines. Clear ablation and few-shot analysis are also provided.

This paper is well-written and introduces a novel direction of transferring vulnerability detection knowledge from Solidity to Vyper, but its innovation and theoretical grounding are limited. With stronger analysis on why MMD-based alignment works; evidence that it truly captures shared semantics rather than superficial similarities; and comparison with recent LLM baselines, the paper could make a more convincing contribution.

**Strengths:**

- This paper tackles a practical and important problem: vulnerability detection for low-resource smart contract languages.
- The idea of leveraging SlithIR as the intermediate representation to bridge Vyper and Solidity is interesting and intuitive.
- The few-shot analysis provides significant evidence to demonstrate the effectiveness of the proposed transfer learning method.
- The presentation is clean, the problem is well motivated, and the method pipeline is easy to follow.

**Weaknesses:**

- The novelty of this paper is limited. The proposed framework combines several well-explored components — a multi-encoder architecture (Transformer + GAT), SlithIR as an intermediate representation, and MMD-based distribution alignment. While the integration is technically coherent, none of these elements are novel by themselves, and the paper does not introduce new algorithmic or theoretical innovations beyond combining existing methods in a new setting.
- The theoretical and technical justification are insufficient. In Stage 1, the model is trained with the MMD objective only, without any discriminative or regularization terms. This design risks representation collapse (where all embeddings converge to a constant vector, trivially minimizing MMD), and a more reasonable approach might jointly optimize the alignment (MMD) and classification objectives to preserve discriminative semantics. However, the paper provides neither empirical discussion nor theoretical support for its current two-stage setup.
- The method assumes that Solidity and Vyper can be aligned in the SlithIR space, but the paper does not analyze or validate this assumption. There is no sufficient evidence demonstrating that the alignment learned by MMD reflects semantic correspondence rather than superficial or noisy correlations. No visualization or feature analysis (e.g., t-SNE) is provided to support the claim of meaningful cross-language alignment.
- Both the training and testing datasets are non-public, and key details such as sample size, labeling quality, and class balance are missing. This limits the reproducibility and external verification of the claimed results.
- The approach and baselines reflect a pre-LLM paradigm. Modern large language models are already capable of directly reasoning about Vyper smart contracts and detecting vulnerabilities without explicit feature alignment. The paper does not compare against such baselines, which weakens both the motivation and the practical relevance of the proposed method.

**Questions:**

- 1. In Stage 1, the model is trained purely with the MMD objective. How do the authors prevent representation collapse problme during this unsupervised alignment? Have the authors considered jointly optimizing the MMD loss and the classification loss (i.e., training Stages 1 and 2 together, or SimCLR) to preserve discriminative semantics?
- 2. The paper would benefit from a clearer theoretical or empirical justification for why minimizing MMD between Solidity and Vyper embeddings improves target-domain generalization, which would strengthen the technical soundness.
- 3. Could the authors provide quantitative or visual evidence (e.g., feature similarity analysis, t-SNE visualization, intra-class variance) that the alignment captures semantic correspondence rather than noise or superficial structure?
- 4. Are there any plan to release the code and data?
- 5. It would strengthen the paper to include a comparison with modern LLMs and explain why explicit MMD-based alignment remains necessary in the current LLM era.

---

### Official Review · Reviewer_G2Z2 · 2025-10-26

**Soundness:** 3
**Presentation:** 3
**Contribution:** 3
**Rating:** 4
**Confidence:** 3

**Summary:**

The paper introduces a novel framework designed to detect vulnerabilities in Vyper smart contracts. Vyper is a new alternative to the commonly used Solidity language. Sol2Vy follows a three-stage approach to utilize Solidity-trained models for Vyper:

- Contracts are converted to SlithIR (intermediate representation) using Slither. Then, a multi-modal encoder trained to minimize MMD between the feature vectors from both languages and extract language-agnostic feature vectors.
- A classification network is trained on labeled Solidity contracts to detect safe or vulnerable instances based on extracted feature vectors.
- Combined network is applied on extracted feature vectors for Vyper contracts.

Experimental results are promising on various vulnerability types compared to traditional techniques and also a baseline deep-learning-based approach.

**Strengths:**

- The problem is critical and the proposed solution is well-motivated. Even though utilizing IRs for cross-language analysis is common, in this instance solely relying on SlithIR is not enough. This motivates the proposed three-stage approach.
- Claims in the paper on the insufficiency of directly using models trained solely on SlithIR or models trained solely on Vyper dataset are supported in experiments.
- Few-shot analysis results are promising for practical use-cases where incorporating a few labeled Vyper contract is possible to improve performance.

**Weaknesses:**

- Technical details especially related to the architecture of the multi-modal encoder are missing. It is not clear to me what the authors mean by a Sequential Encoder or Graph Attention Network. I also don’t see the source code as supplementary material or architecture details in appendix.
- It is not clear how curriculum learning is applied. What defines the complexity of unlabeled contract dataset samples?
- It is not clear how the synthetic dataset for Vyper is created.

**Questions:**

- Could you provide the details of the model architecture and hyperparameters during training?
- Why do you think balanced few shot examples (e.g. 4S+4V compared 8S or 8V) produce better results? What about other proportions of samples from safe and vulnerable contracts?

---

### Official Review · Reviewer_b3qn · 2025-10-31

**Soundness:** 1
**Presentation:** 2
**Contribution:** 2
**Rating:** 2
**Confidence:** 4

**Summary:**

This paper introduces Sol2Vy, a framework for detecting vulnerabilities in Vyper smart contracts by leveraging Solidity-trained models, thereby avoiding the need for labeled Vyper datasets. The approach uses SlithIR, a language-agnostic intermediate representation, and employs Maximum Mean Discrepancy (MMD) loss to align feature distributions between Solidity and Vyper representations. The pipeline consists of three stages: representation extraction, feature alignment, and supervised learning on Solidity data, followed by zero-shot application to Vyper.

**Strengths:**

- The paper tackles the cross-language security gap in smart contract analysis, specifically between Solidity and Vyper. This is an underexplored but practically relevant challenge in blockchain security.

- The overall structure of the paper is clear, and the motivation and methodology are presented in a logical manner.

**Weaknesses:**

** Limited novelty**
 The idea of aligning source and target language representations in latent space using domain adaptation (e.g., MMD) is well-established. The paper does not sufficiently highlight what is conceptually new beyond applying this to Solidity–Vyper transfer. Since SlithIR already exists as a representation layer, the novelty of the contribution appears incremental.


**Unclear methodological details & poor reproducibility**

   (a) Referring to sequence and graph representations as “multi-modal” is conceptually inaccurate, as these are typically considered multi-view rather than multi-modal.

   (b) Missing architecture details. Critical implementation details such as the number of Transformer layers/heads, hidden dimensions, and GAT configurations are missing, making replication difficult.

   (c) Insufficient justification of kernel choice. Although some hyperparameter tuning is mentioned, the rationale for using a linear + RBF kernel combination in MMD, and the search range for parameters, are not justified.


** Insufficient baselines and ablations**

   (a) Missing control baseline. The paper lacks a direct comparison to a model trained on SlithIR without MMD alignment. Such an ablation is essential to demonstrate that cross-language alignment indeed contributes to performance improvements.

   (b) No comparison with pretrained code models.  Pretrained models like CodeBERT, GraphCodeBERT, or even large language models have strong cross-language generalization capabilities. Evaluating against these zero- or few-shot baselines is necessary to position the contribution.

   (c) Missing cross-language representation baselines. Other cross-language program representation learning methods (e.g., code translation alignment, contrastive pretraining) should also be included for fairness.


** Limited discussion of applicability and generalization**

The current evaluation focuses on three vulnerability types. It is unclear how well the approach generalizes to semantic or logic-based vulnerabilities, such as reentrancy across complex control flows or access control flaws. A broader discussion of this limitation is missing.

**Questions:**

Q1: Have the authors considered leveraging pretrained language models as encoders within the framework rather than training from scratch?

Q2: What is the runtime overhead compared to existing static analysis tools?

---

### Official Review · Reviewer_gY2Y · 2025-10-31

**Soundness:** 2
**Presentation:** 3
**Contribution:** 2
**Rating:** 2
**Confidence:** 4

**Summary:**

This paper presents Sol2Vy, a method of using pre-trained models on solidity with large labeled data to provide vulnerability detection in Vyper smart contracts. The main idea is to map Solidity smart contract and Viper smart contracts to SlithIR, a common representation for Ethereum smart contract languages (originally proposed in Slither (Feist et al 2019) and then design an encoder to capture the language-agnostic semantics in the both Solidity and Viper smart contracts.

**Strengths:**

The idea of transforming Solidity smart contract and Viper smart contracts to SlithIR, a common representation for Ethereum smart contract languages is interesting. If the transformation of both smart contract datasets will be made available in public domain, it may help the smart contract vulnerability detection research field.

The idea of design an encoder to be trained on the two datasets transformed into the common format SlithIR is well motivated: to capture the language-agnostic semantics in both Solidity and Viper smart contracts.

The experiments are performed to compare the proposal approach with two existing approaches (Slither and Mythril) and the model that trained and tested only in Vyper which is claimed to have less labeled data.

**Weaknesses:**

This paper can benefit from a number of improvements:

(1) The idea of design an encoder to be trained on the two datasets transformed into the common format SlithIR is well motivated: to capture the language-agnostic semantics in both Solidity and Viper smart contracts.  However, the paper could articulate the technical novelty in the encoder training process, such as different ways/algorithms of training an encoder and why the proposed approach will be better.

(2) The paper could benefit by clearly defining the concept of the language-agnostic semantics in both Solidity and Viper smart contracts with examples. Given this is the most important technical highlight of the paper, more elaboration on why your proposed approach to train such decoder will provide better learning results on the language-agnostic semantics in both Solidity and Viper smart contracts.

(3) The experiments are somewhat weak and need to provide more in-depth experiments and ablation studies.

(3.1) The experimental results in Table 1 should be more clearly discussed. For example, the comparison with the Same-Language model (trained &tested on Vyper) is not a fair comparison since Sol2Vy utilizes additional solidity smart contracts and Vyper smart contracts in getting language-agnostic semantic knowledges. Hence, a further study on utilizing this decoder to train the Vyper dataset should be provided.

(3.2) Similarly, when comparing with Nythril and Slither, it might be useful to also run both on top of the Sol2Vy encoder so the readers can understand the role of this encoder in improving the performance of the vulnerability detection independently of the solidity based supervised training in your step 2.

(3.3) In Table 2 with few shot settings, the 12-shot setting performs the best compared to 16-shots and 20-shots. Hence, the paper would benefit from a detailed discussion on how the shots are selected and the quality of the shots. When randomly choosing 12 shots from the 20-shot setting, would the proposed approach performs consistently?

(3.4) How does the heat map in Figure 2 help understand/illustrate the results of Table 1 and Table 2.

**Questions:**

The paper is an application-driven paper. The authors may want to discuss the broader impact of the proposed approach.

Additional questions see the weakness section.

---

### Note · Authors · 2025-12-03

**Comment:**

We would like to thank all the anonymous reviewers for their time in reviewing this paper. We will carefully use the suggestions to improve this work in the future.

**Withdrawal Confirmation:**

I have read and agree with the venue's withdrawal policy on behalf of myself and my co-authors.